# COVID-19 Victimization Experience and College Students’ Mobile Phone Addiction: A Moderated Mediation Effect of Future Anxiety and Mindfulness

**DOI:** 10.3390/ijerph19137578

**Published:** 2022-06-21

**Authors:** Lili Chen, Jun Li, Jianhao Huang

**Affiliations:** 1School of Information Engineering, Hainan Technology and Business College, Haikou 570203, China; lilichen.edu.hgs@gmail.com; 2Chinese International College, Dhurakij Pundit University, Bangkok 10210, Thailand; lijun.edu.ma@foxmail.com

**Keywords:** COVID-19 victimization experience, mobile phone addiction, future anxiety, mindfulness, college students

## Abstract

This study proposed a moderated mediation model to investigate the association between COVID-19 victimization experience and mobile phone addiction, the mediating role of future anxiety, and the moderating role of mindfulness. This study employed the COVID-19 victimization experience scale, the mobile phone addiction scale, a future anxiety scale, and a mindfulness scale in a survey study among Chinese college students; 840 valid questionnaires were received. The reliability and confirmatory factor analysis results showed that all four scales had good reliability and validity. Bootstrap results demonstrated that COVID-19 victimization experience significantly predicted mobile phone addiction in college students (B = 0.202, LLCI = 0.136, ULCI = 0.268). Future anxiety fully mediated the association between COVID-19 victimization experience and mobile phone addiction (B = 0.178, LLCI = 0.136, ULCI = 0.222). Mindfulness moderated the effect of COVID-19 victimization experience on the college students’ future anxiety (B = 0.159, LLCI = 0.007, ULCI = 0.054). A higher level of mindfulness was more likely than a lower level of mindfulness to attenuate the effect of COVID-19 victimization experience on the college students’ future anxiety. These findings broaden our understanding regarding the association between COVID-19 victimization experience and mobile phone addiction and the moderating role of mindfulness.

## 1. Introduction

COVID-19, which broke out in late 2019, rapidly developed into a public health emergency that drew international attention, escalating into a pandemic by March 2020 [1]. During the COVID-19 pandemic, smartphones play an essential role in communication, access, and information sharing [2]. Past findings suggest that the COVID-19 pandemic has increased the frequency and dependence of individuals on smartphones [3,4] which might lead to mobile phone addiction among college students [5]. A recent study reported an increased prevalence of mobile phone addiction among students during the COVID-19 pandemic [6]. Studies have also reported that mobile phone addiction negatively affects college students’ academic performance and mental health [7,8]. Therefore, it is crucial to explore the critical influences on mobile phone addiction among college students during the COVID-19 pandemic.

Mobile phone addiction refers to the loss of self-control resulting from the excessive usage of mobile phones and leads to difficulties in daily life [9]. Research has shown that the fear of COVID-19 and anxiety regarding COVID-19 infection are identified as significant positive predictors of smartphone addiction [10,11]. Another study demonstrated that trauma exerted a significant predictive effect on mobile phone addiction among college students [12]. Therefore, the traumatic experience brought by COVID-19 is likely a significant predictor of mobile phone addiction in college students. However, the potential mechanistic implications for this process remain largely unknown and need further exploration. Existing research has found that COVID-19 perceived threat increases future anxiety [13] and that future anxiety predicted mobile phone addiction [14]. Additional findings have reported that mindfulness moderates the relationship between trauma and anxiety symptoms in college students [15]. To the best of our knowledge, limited studies have investigated the mediating effect of future anxiety in the association between COVID-19 victimization experience and mobile phone addiction and the moderating role of mindfulness in the mediating effect. Therefore, the present study attempted to fill the gap by examining the mediating role of future anxiety and the moderating role of mindfulness in the association between COVID-19 victimization experience and mobile phone addiction. The findings of this empirical study can help elucidate crucial factors affecting college students’ mobile phone addiction, thereby improving our understanding on the potential association between COVID-19 victimization experience and college students’ mobile phone addiction and the moderating role of mindfulness. The study findings can provide new directions for educators for effectively reducing the risk of mobile phone addiction among college students.

### 1.1. Social Cognitive Theory

Social cognitive theory (SCT), proposed by Bandura [16], argues that intraindividual factors, external environment, and behavior interact with each other. Studies on mobile phone addiction have applied SCT [17,18,19]. In particular, Lian et al. [17] considered parenting style as an environmental factor and virtue as a personal factor based on SCT. They determined that virtue reduced mobile phone addiction and that parenting style significantly affected mobile phone addiction. On the basis of SCT, Kara et al. [18] regarded the duration of daily smartphone usage as an environmental factor and regarded loneliness and general anxiety as personal factors; their findings revealed that the duration of daily smartphone usage, loneliness, and anxiety all significantly affected mobile phone addiction. Cheng et al. [19] considered the parent–child relationship as an environmental factor and loneliness and self-efficacy as personal factors based on SCT. Their results revealed that an improvement in the parent–child relationship reduced loneliness and mobile phone addiction and an improvement in self-efficacy reduced mobile phone addiction. Therefore, this study regarded COVID-19 victimization experience as an environmental factor and future anxiety and mindfulness as individual factors based on SCT to investigate the association between COVID-19 victimization experience and mobile phone addiction. In addition, the mediating role of future anxiety and the moderating role of mindfulness in the association were examined.

### 1.2. COVID-19 Victimization Experience and Mobile Addiction

Recent studies have shown a significant increase in the use of smartphones to access social networks, the internet, and entertainment applications during the COVID-19 pandemic [20,21]. Increased smartphone use may contribute to mobile phone addiction during the COVID-19 pandemic [22]. Currently, several studies have identified factors causing mobile phone addiction during the COVID-19 pandemic. For example, household financial decline due to COVID-19, peer phubbing, interpersonal alienation, loneliness, escape motivation, perceived stress, and social cynicism have been reported to be significant positive predictors of mobile phone addiction [23,24,25,26,27,28]. Kayis et al. [10] conducted a questionnaire survey among 773 adults and determined that the fear of COVID-19 had a significant positive predictive effect on mobile phone addiction. Another empirical study including 550 adults demonstrated that the burden caused by COVID-19 exerted a significant positive prediction effect on addictive social media use behavior [29]. Liang et al. [12] conducted a questionnaire survey among 263 college students and determined that trauma exerted a significant positive predictive effect on mobile phone addiction. On the basis of the aforementioned findings. Hypothesis 1 was proposed as follows: 

**Hypothesis** **1.***COVID-19 victimization experience would exert a significant positive prediction effect on mobile phone addiction among college students*.

### 1.3. Mediating Role of Future Anxiety

Future anxiety refers to a state in which an individual feels uncertainty, fear, and worry regarding the prospect of adverse changes during the future [30,31]. Studies have found that the COVID-19 pandemic has considerably disrupted individuals’ lives, leading to uncertainty regarding the future [13,32]. The COVID-19 pandemic increased anxiety regarding the future [33,34,35]. Additionally, the COVID-19 outbreak has exacerbated economic and social problems, including unemployment and economic recession, generating anticipatory fears that in turn increase anxiety regarding the future [13]. Therefore, COVID-19 victimization experience might exert a positive predictive effect on future anxiety. In addition, online social anxiety was reported to significantly and positively predict mobile phone addiction during the COVID-19 pandemic [4]. Annoni et al. [36] conducted a questionnaire survey among 240 adults and reported that social anxiety significantly and positively predicted mobile phone addiction. In contrast to the concept of anxiety, which focuses on the immediate future, future anxiety involves a more remote personal future [30]. Przepiorka et al. [14] conducted a questionnaire survey among 478 students and determined that future anxiety significantly and positively predicted mobile phone addiction.

Previous findings have suggested that future anxiety often plays an essential mediating role. For example, Paredes et al. [13] found that future anxiety mediated the association between the perceived threat of COVID-19 and mental well-being. Przepiorka et al. [14] reported that future anxiety mediated the correlation between procrastination and mobile phone addiction. Therefore, Hypothesis 2 was proposed as follows: 

**Hypothesis** **2.***Future anxiety would mediate the association between COVID-19 victimization experience and mobile phone addiction among college students*.

### 1.4. Moderating Role of Mindfulness

Mindfulness is defined as a process that involves attention, awareness, and open-minded acceptance of the present moment [37]. A study suggested that individuals with high levels of mindfulness were more receptive to the COVID-19 pandemic and experienced less stress and anxiety [38]. An increase in mindfulness is associated with a decrease in other negative emotions in college students, including anxiety, indicating that high levels of mindfulness among college students can reduce their anxiety [38,39,40]. In addition, a study reported that individuals with lower levels of mindfulness had higher fear of COVID-19 [41] and more severe anxiety symptoms [42]. When individuals have low levels of mindfulness, they cannot focus their attention on the present moment and are highly affected by the outside environment; thus, they tend to view things as developing beyond their control, which can lead to feelings of anxiety and depression [43]. In their empirical study including 2336 college students, Tubbs et al. [15] observed that mindfulness played a moderating role in the relationship between trauma and anxiety symptoms. In particular, high levels of mindfulness attenuated the effect of trauma on anxiety symptoms.

The aforementioned findings indicate that mindfulness may alleviate the effects of COVID-19 victimization experience on future anxiety among college students. Therefore, Hypothesis 3 was proposed as follows: 

**Hypothesis** **3.***Mindfulness would play a moderating role in the association between COVID-19 victimization experience and future anxiety among college students*.

### 1.5. The Present Study

The present study constructed a moderated mediation model (Figure 1), and the research hypotheses were as follows: (1) COVID-19 victimization experience would exert a significant positive prediction effect on mobile phone addiction among college students; (2) future anxiety would mediate the association between COVID-19 victimization experience and mobile phone addiction among college students; and (3) mindfulness would play a moderating role in the association between COVID-19 victimization experience and future anxiety among college students.

## 2. Materials and Methods

### 2.1. Ethics Approval

The Ethics Committee of the Hainan Technology and Business College (HGS-2022-02) approved the present study. It was conducted following the ethical standards required for conducting human research, in accordance with the fundamental principles in the Declaration of Helsinki [44]. Participants were asked to indicate whether they wished to participate in the present study before completing the survey. The data in this study were collected and analyzed anonymously.

### 2.2. Participants and Procedure

Convenience sampling was used in this study, and data were collected from 27–30 October 2021. Participants were recruited from two universities in China’s Yunnan and Hainan provinces. The following operations were conducted to control potential information bias effectively: (a) The head teachers of classes were responsible for distributing the questionnaires. Before recruitment, professional training was provided to them to explain the questionnaire entries and the recruiting criteria (participants should be college students who were interested in the topic of mobile phone addiction and voluntarily engaged in the present study). Furthermore, they were required to introduce the questionnaire information to the participants before filling it out. (b) All participants were also informed by head teachers of the purpose of the study and the confidentiality agreement (the questionnaire was submitted anonymously, and the data were processed anonymously) and that they could refuse to participate or withdraw from the study at any time if they had any doubts during the process of filling out the questionnaire. The process of filling out the questionnaire was completed with the help of the head teachers.

After participants gave written consent, questionnaires were distributed through Questionnaire Star (www.wjx.cn), a widely used online questionnaire app in China. They could scan a QR code (a 2D barcode containing a link to the online questionnaire) or click on the questionnaire link to fill in the online questionnaire directly. The average response period was 9 min. In total, 167 samples with response periods shorter than 3 min or longer than 15 min or missing values were excluded. A total of 840 valid questionnaires were returned, with a return rate of 83.4%.

### 2.3. Instruments

#### 2.3.1. COVID-19 Victimization Experience Scale

The COVID-19 victimization experience scale developed by Yang et al. [45,46] was used to measure trauma resulting from the COVID-19 pandemic experience among college students. The eight-question scale has two dimensions: catastrophic cognition (e.g., I do not think anyone has had a worse experience than me) and trauma symptoms (e.g., I am often worried about COVID-19 infection). The responses are rated on a 5-point Likert scale, ranging from 1 (strongly disagree) to 5 (strongly agree), with higher scores indicating a higher level of traumatic experience of the COVID-19 pandemic.

#### 2.3.2. Mobile Phone Addiction Scale

The mobile phone addiction scale developed by Leung [47] was used in this study to measure the level of mobile phone addiction in college students. The 17-question scale comprises four dimensions: inability to control craving (e.g., You have attempted to spend less time on your mobile phone but are unable to), feeling anxious and lost (e.g., You feel anxious if you have not checked messages or switched on your mobile phone for some time), withdrawal/escape (e.g., You have used your mobile phone to talk to others when you were feeling isolated), and productivity loss (e.g., Your productivity has decreased as a direct result of the time you spend on the mobile phone). The responses are rated on a 5-point Likert scale ranging from 1 (not at all) to 5 (always), with higher scores indicating higher levels of mobile phone addiction.

#### 2.3.3. Future Anxiety Scale

This study used the Dark Future Scale developed by Zaleski et al. [48] to measure the degree of future anxiety in the college students. This scale consists of five questions related to one dimension (e.g., I am afraid that changes in the economic and political situation will threaten my future). The responses are rated on a 7-point Likert scale, with scores ranging from 0 (decidedly false) to 6 (decidedly true), and a higher score indicates a higher degree of future anxiety.

#### 2.3.4. Mindfulness Scale

Greco et al. [49] developed the unidimensional Child and Adolescent Mindfulness Measure (CAMM), which was revised and validated in the Chinese adolescent population by Liu et al. [50]. The revised scale with two dimensions exhibited satisfactory reliability and validity. Therefore, the revised Chinese version of the scale was used in this study to measure the level of mindfulness in the Chinese college students. The two dimensions are awareness and nonjudgment (e.g., I get upset with myself for having feelings that do not make sense) and acceptance (e.g., I think that some of my feelings are bad and that I should not have them). The responses for the 10 questions are rated on a 5-point Likert scale ranging from 0 (never) to 4 (always). All items in the scale were reverse scored, with higher scores indicating higher levels of mindfulness among the college students.

#### 2.3.5. Data Analysis

SPSS (version 21.0, IBM, Armonk, NY, USA) was used in this study to conduct the following data analysis, and the statistical significance standard was set at *p* < 0.05 throughout the data analysis:Descriptive statistics were performed on the participants’ background information, indicating the proportion of participants’ composition.Descriptive statistics and correlation analysis were performed for each variable, with descriptive statistics reflecting each variable’s means and standard deviations. Pearson’s correlation test was used to analyze the correlation between each variable and dimension. When the correlation coefficient should be less than 0.7 [51], indicating no collinearity problem in all variables, then the regression analysis could be performed.The value of Cronbach’s α tested the reliability of each scale. Values greater than 0.7 indicated the excellent reliability of the measurement instrument [52].The common method variance (CMV) was tested as well as the unrotated factor analysis by Harman’s One-Factor Test; when the Kaiser–Meyer–Olkin (KMO) was greater than 0.8, Bartlett test of sphericity reached significance. The explanatory power of the first factor should not exceed the marginal value of 50% [53], indicating that the CMV problem is not significant.The mediating effect of future anxiety was tested using model 4 of the Hayes PROCESS plug-in with COVID-19 victimization experience as the independent variable and mobile phone addiction as the dependent variable; mindfulness was added as the moderating variable in the model 7 of the Hayes PROCESS plug-in to test the moderated mediation model. Additionally, the Bootstrap confidence interval was set to 95%, and the sample size was set to 5000. The CI (from the lower limit of confidence interval (LLCI) to the upper limit of confidence interval (ULCI)) of each path coefficient should not contain 0, meaning a significant effect [54].

In addition, confirmatory factor analysis (CFA) was conducted with AMOS (version 21.0, IBM, Armonk, NY, USA) and met the criteria to indicate good validity of the measurement model.

The factor loadings of the measurement models were tested with the criterion of greater than 0.5. The values of composite reliability (CR) were tested with the criterion of greater than 0.7. Moreover, the values of average variance extracted (AVE) were tested with the criterion of greater than 0.5. All the above indicators were satisfied, indicating the good convergent validity of the scales [51].The fitness of the measurement model was tested using the following essential indicators. The Chi-square value should not be significant (*p* > 0.05). However, considering the sensitivity of the Chi-square value to the large sample size (when the sample size is large, the Chi-square value can easily reach significance), it was not reported in this study and other indicators were tested and referred to [55]. Namely, root mean square residual (RMR) < 0.08; standardized RMR (SRMR) = < 0.08; comparative fit index (CFI) > 0.85; goodness-of-fit index (GFI) > 0.85; normed fit index (NFI) > 0.85; Tucker–Lewis index (TLI) > 0.80 and incremental fit index (IFI) > 0.85 [56]. If the above criteria were met, the measured model fitness was acceptable.The square root of AVE was performed to assess the discriminant validity of each dimension of the measurement model, with the criterion of the square root of AVE greater than the correlation coefficient in each dimension [57].

## 3. Results

### 3.1. Participants’ Composition

A total of 840 participants were enrolled in this study, 275 (32.7%) were male students and 565 (67.3%) were female students, 438 (52.1%) were general undergraduate students and 402 (47.9%) were vocational undergraduate students, 280 (33.3%) resided in urban areas and 560 (66.7%) in rural areas, 198 (23.6%) were the only children, and 642 (76.4%) had siblings. The sample included first- to fourth-year university students aged from 18 to 23 years.

### 3.2. Measurement Model

#### 3.2.1. COVID-19 Victimization Experience Scale

The Cronbach’s α value of the scale was 0.886 (>0.7), indicating satisfactory reliability [52]. The results of the CFA are listed in Table 1. The CFA had standardized factor loadings of 0.622–0.810 (both > 0.5), indicating the satisfactory validity of the scale [58]. Composite reliability (CR) values were 0.844 and 0.826 (both > 0.7), and average variance extracted (AVE) values were 0.576 and 0.546 (both > 0.5), indicating the high convergent validity of the scale [51]. The model fit indices of the measurement model were as follows: root mean square residual (RMR) = 0.079, standardized RMR (SRMR) = 0.063, comparative fit index (CFI) = 0.880, goodness-of-fit index (GFI) = 0.873, normed fit index (NFI) = 0.875, Tucker–Lewis index (TLI) = 0.823, and incremental fit index (IFI) = 0.880. The values indicated an acceptable fit of the measurement model to the observed data [56].

#### 3.2.2. Mobile Phone Addiction Scale

The Cronbach’s α value of the scale was 0.904 (>0.7), indicating satisfactory reliability [52]. Table 2 presents the results of CFA. The standardized factor loadings ranged from 0.519 to 0.863, indicating the satisfactory validity of the scale [58]. CR values ranged from 0.738 to 0.848 (>0.7) [51], and AVE values ranged from 0.445 to 0.588; According to Fornell and Laecker [57], even if the AVE is less than 0.5, the convergent validity of the scale is still acceptable under the conditions of the CR value meeting the criteria (greater than 0.6). The model fit indices of the measurement model were as follows: RMR = 0.077, SRMR = 0.065, CFI = 0.894, GFI = 0.893, NFI = 0.879, TLI = 0.873, and IFI = 0.895. These values indicated an acceptable fit of the measurement model to the observed data [56].

#### 3.2.3. Future Anxiety Scale

The Cronbach’s α value of the scale was 0.897 (>0.7), indicating satisfactory reliability [52]. Table 3 presents the results of CFA. The standardized factor loadings ranged from 0.692 to 0.874 (>0.5), indicating satisfactory validity [58]. The CR value was 0.897 (>0.7), and the AVE value was 0.638 (>0.5), indicating satisfactory convergent validity [51]. The model fit index values were as follows: RMR = 0.078, SRMR = 0.032, CFI = 0.973, GFI = 0.966, NFI = 0.971, TLI = 0.946, and IFI = 0.973. These values indicated an acceptable fit of the measurement model to the observed data [56].

#### 3.2.4. Mindfulness Scale

The Cronbach’s α of the scale was 0.823 (>0.7), indicating satisfactory reliability [52]. The results of CFA are listed in Table 4. The standardized factor loadings ranged from 0.524 to 0.820 (both > 0.5), suggesting the scale had high validity [58]. The CR values were 0.819 and 0.798 (both > 0.7) [51], and AVE values were 0.436 and 0.502. Even if the AVE is less than 0.5, the scale’s convergent validity is still allowed if the CR value meets the criteria of greater than 0.6 [57]. The model fit indices were as follows: RMR = 0.060, SRMR = 0.064, CFI = 0.892, GFI = 0.920, NFI = 0.883, TLI = 0.858, and IFI = 0.893. These values indicated the acceptable fit of the measurement model to the observed data [56].

### 3.3. Discriminant Validity 

The square root of AVE was performed to rigorously examine the discriminant validity of all the scales in the present study. The criterion was that the square root value of AVE was greater than the correlation coefficient in each dimension [57]. As shown in Table 5, the results met the criteria for assessing discriminant validity, indicating that each scale in this study had good discriminant validity.

### 3.4. Common Method Variance (CMV) Test

Harman’s one-factor test was performed to examine CMV. Unrotated factor analysis revealed that the Kaiser–Meyer–Olkin (KMO) value was 0.920 (>0.8), and the results of Bartlett’s test of sphericity reached significance (*p* < 0.001). The analysis yielded eight factors, and the explanatory power of the first factor was 27.126%, which did not exceed the critical value of 50% [53], indicating that no severe CMV problem existed in this study.

### 3.5. Descriptive Statistics and Correlation Analysis

Table 6 presents the descriptive statistics for the following four variables: COVID-19 victimization experience, mobile phone addiction, future anxiety, and mindfulness. The results of correlation analysis revealed that COVID-19 victimization experience was positively correlated with mobile phone addiction (r = 0.240, *p* < 0.001), positively correlated with future anxiety (r = 0.579, *p* < 0.001), and negatively correlated with mindfulness (r = −0.244, *p* < 0.001). Future anxiety and mobile phone addiction were positively correlated (r = 0.382, *p* < 0.001). Mindfulness was negatively correlated with mobile phone addiction (r = −0.625, *p* < 0.001) and future anxiety (r = −0.392, *p* < 0.001). The absolute values of correlation coefficients between any two variables were <0.7, and no collinearity problem was observed [51].

### 3.6. Mediating Role of Future Anxiety

The mediating effect of future anxiety was examined using Model 4 of the process. Table 7 presents the results. In Model 1, COVID-19 victimization experience significantly and positively predicted mobile phone addiction (B = 0.202, *p* < 0.001). In Model 2, COVID-19 victimization experience significantly and positively predicted future anxiety (B = 0.931, *p* < 0.001). When added as a mediating variable in Model 3, future anxiety significantly and positively predicted mobile phone addiction (B = 0.191, *p* < 0.001); however, the predictive effect of COVID-19 victimization experience on mobile phone addiction was nonsignificant (B = 0.024, *p* > 0.05). Future anxiety fully mediated the effect of COVID-19 victimization experience on mobile phone addiction. Furthermore, this study investigated the mediating effect of future anxiety by using bias-corrected nonparametric percentile bootstrapping, and the indirect effect value was 0.178 with 95% CI values ranging from 0.136 (lower limit of confidence interval, abbreviation: LLCI) to 0.222 (upper limit of confidence interval, abbreviation: ULCI), which does not contain 0, indicating the mediating effect of future anxiety. The direct effect value was 0.024, with 95% CI values ranging from −0.050 (LLCI) to 0.095 (ULCI), including 0, indicating the full mediating effect of future anxiety; the mediating effect accounted for 88.119% of the total effect.

### 3.7. Moderating Role of Mindfulness

Model 7 was applied to examine whether mindfulness moderated the first half of the mediated model (Table 8). A significant positive predictive effect of COVID-19 victimization experience on future anxiety (B = 0.827, *p* < 0.001) was observed in Model 1. The interaction of COVID-19 victimization experience with mindfulness had a significant predictive effect on future anxiety (B = 0.159, *p* < 0.01), indicating that mindfulness plays a moderating role. The predictive effect of COVID-19 victimization experience on mobile phone addiction in Model 2 was nonsignificant (B = 0.024, *p* > 0.05); however, future anxiety was a significant positive predictor of mobile phone addiction (B = 0.191, *p* < 0.001). The results were verified through bias-corrected nonparametric percentile bootstrapping. The index of moderated mediation was 0.030 (LLCI = 0.007 and ULCI = 0.054), with the 95% CI not including 0, indicating that the moderated mediation model holds [59]. To determine the moderating effect of mindfulness, a simple slope analysis was performed, and two graphs were plotted for the moderating effect for the two groups of mindfulness in Figure 2: (1) high mindfulness (mean + 1 SD) and (2) low mindfulness (mean − 1 SD). The results revealed that the effect of COVID-19 victimization experience on future anxiety was stronger in the college students with higher levels of mindfulness (simple slope = 0.922, t = 16.264, *p* < 0.001) than in those with lower levels of mindfulness (simple slope = 0.732, t = 12.982, *p* < 0.001). In other words, a higher level of mindfulness was more likely than a lower level of mindfulness to attenuate the effect of COVID-19 victimization experience on future anxiety in the college students.

## 4. Discussion

### 4.1. Theoretical Contributions

First, the current findings indicated that COVID-19 victimization experience had a significant predictive effect on mobile phone addiction in the surveyed college students. This finding is similar to that of a previous study [12] that demonstrated a significant predictive effect of trauma on mobile phone addiction. Because individuals spent more time at home during the COVID-19 pandemic and had more free time, their smartphone use increased for online activities such as browsing social media, watching movies or series, and listening to music [60]. Moreover, increased frequency of smartphone use [61,62] is a critical sign of mobile phone addiction and may lead to the development of mobile phone addiction in some individuals. In addition, during the pandemic, the frequency of smartphone use increased for accessing information related to the COVID-19 pandemic as well as for study and work [63]. Compulsive use and regular checking of information are common symptoms of mobile phone addiction [64,65]. Therefore, COVID-19 victimization experience can lead to mobile phone addiction problems in college students.

Second, the findings revealed that future anxiety mediated the association between COVID-19 victimization experience and mobile phone addiction in the surveyed college students. COVID-19 victimization experience indirectly affected the college students’ mobile phone addiction through future anxiety. The results are in agreement with those of a previous study that indicated that the outbreak of the COVID-19 pandemic exacerbated economic and social problems and thus increased anxiety regarding the future [13]. Moreover, studies have revealed that increased future anxiety increases the risk of mobile phone addiction [14] and that anxiety is significantly higher in individuals who experienced a traumatic event [66]. When individuals experience psychological problems in the real world, they may use virtual networks or smartphones to escape negative emotions [67]. Individuals with COVID-19 victimization experience may have negative thoughts regarding the future and feel anxious regarding their situation; thus, they tend to use their phones frequently to escape reality. Moreover, mobile phone use may ease their negative emotions regarding tasks to be completed and decisions to be made in the future [14]. Zis et al. [68] also revealed that the COVID-19 pandemic negatively affected the mental health of college students. This study further broadens the results of the aforementioned studies.

Third, the findings indicated that mindfulness moderated the association between COVID-19 victimization experience and future anxiety in the surveyed college students. This finding is similar to that of a previous study [15] demonstrating that mindfulness attenuated the effects of trauma on college students’ anxiety. In particular, higher levels of mindfulness were more likely than lower levels of mindfulness to attenuate the effects of COVID-19 victimization experience on future anxiety in the college students (Figure 2). Mindfulness can help individuals direct their attention to current experiences [43] by inhibiting rumination regarding past and future experiences [69]. Mindfulness is associated with positive well-being and negatively associated with depression and anxiety and can be employed for treating many psychological disorders including anxiety and depression [43]. In addition, a unique attribute of mindfulness is the nonjudgmental awareness of and focus on one’s experience in the present moment [43]. Many anxiety disorders are characterized by concern and stress related to future events [70]. Mindfulness provides a present-oriented focus. Therefore, mindfulness may be uniquely beneficial for those with anxiety following a traumatic event because it shifts an individual’s attention from negative traumatic experiences and anxiety regarding future events to the present. One study revealed that mindfulness reduced psychological distress and enhanced well-being in adults during the COVID-19 pandemic [71]. Moreover, mindfulness can enhance resilience to trauma [72] and reduce anxiety [73,74]. Overall, the present results indicated that mindfulness attenuated the effects of COVID-19 victimization experience on future anxiety in college students.

### 4.2. Practical Contributions

This study makes some valuable practical contributions. First, college teachers should guide students to appropriately face COVID-19 victimization experience and guide them to develop an appropriate understanding of the crisis. Second, colleges and universities can hold lectures on mental health or conduct activities to alleviate students’ anxiety regarding the future. Third, colleges and universities can integrate mindfulness training into mental health courses to improve students’ mindfulness ability.

## 5. Limitations and Future Research Directions

This study has some limitations. First, because this study was cross-sectional, it could not determine causal relationships between variables. Therefore, a longitudinal or experimental study should be conducted in the future. Second, Chinese college students were included as participants in this study. Therefore, future cross-cultural studies should be conducted to compare differences in COVID-19 victimization experience on mobile phone addiction between Chinese and Western college students. Third, the convenience sampling method used in this study might have inferential limitations; therefore, the sampling method could be improved, and the sampling range of the sample could be expanded in future studies. Fourth, this study used a self-report questionnaire to conduct the survey, qualitative interviews could be included in future studies to more effectively investigate the effect of COVID-19 victimization experience on college students’ mobile phone addiction. Fifth, two of the AVE values in the present study were low. Therefore, future studies could design a pretest survey of the sample before the formal test to improve or delete the scale items by using item analysis and exploratory factor analysis so that each item of the scale could obtain a high factor loading.

## 6. Conclusions

This study demonstrated that COVID-19 victimization experience exerted a significant predictive effect on college students’ mobile phone addiction, and the effect was fully mediated by future anxiety. Moreover, mindfulness was determined to play a moderating role in the association between COVID-19 victimization experience and future anxiety. In particular, a higher level of mindfulness was more likely than a lower level of mindfulness to attenuate the effect of COVID-19 victimization experience on future anxiety in college students.

## Figures and Tables

**Figure 1 ijerph-19-07578-f001:**
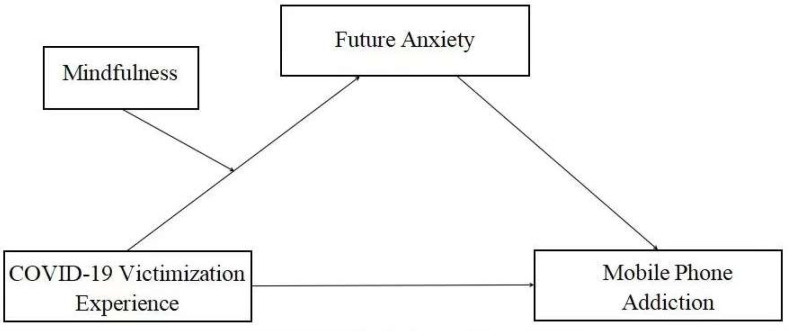
Moderated mediation model.

**Figure 2 ijerph-19-07578-f002:**
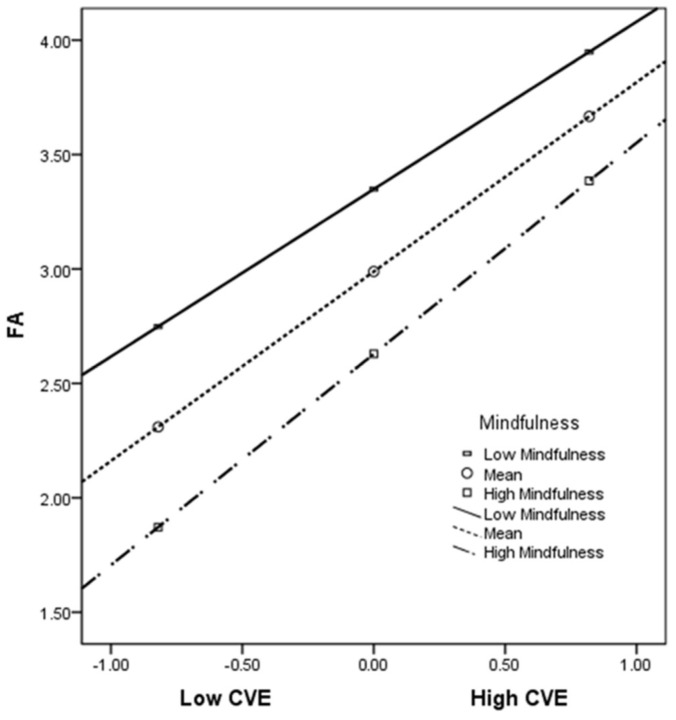
Moderating effect of mindfulness. CVE, COVID-19 victimization experience; FA, future anxiety.

**Table 1 ijerph-19-07578-t001:** Confirmatory factor analysis of the COVID-19 victimization experience scale.

Dimension	Item	FL	CR	AVE
Catastrophic cognition	I do not think anyone has had a worse experience than me	0.686	0.844	0.576
	I think that what happened to me was the worst	0.761		
	I frequently think about how bad things have become	0.772		
	I frequently think about how terrible what had happened to me is	0.810		
Trauma symptoms	My body often feels tense	0.708	0.826	0.546
	I am often worried about becoming infected	0.622		
	I frequently cannot sleep	0.805		
	My mood is always fluctuating	0.805		

FL = standardized factor loading; CR = composite reliability; AVE = average variance extracted.

**Table 2 ijerph-19-07578-t002:** Confirmatory factor analysis of the mobile phone addiction scale.

Dimension	Item	FL	CR	AVE
Inability to control craving	1. Your friends and family complained about your use of the mobile phone	0.684	0.848	0.445
	2. You have been told that you spend too much time on your mobile phone	0.692		
	3. You have tried to hide from others how much time you spend on your mobile phone	0.611		
	4. You have received mobile phone bills you could not afford to pay	0.574		
	5. You find yourself engaged on the mobile phone for longer period of time than intended	0.736		
	6. You have attempted to spend less time on your mobile phone but are unable to	0.732		
	7. You can never spend enough time on your mobile phone	0.621		
Feeling anxious and lost	1. When out of range for some time, you become preoccupied with the thought of missing a call	0.663	0.843	0.523
	2. You find it difficult to switch off your mobile phone	0.772		
	3. You feel anxious if you have not checked for messages or switched on your mobile phone for some time	0.834		
	4. You feel lost without your mobile phone	0.784		
	5. If you do not have a mobile phone, your friends would find it hard to get in touch with you	0.519		
Withdrawal/escape	1. You have used your mobile phone to talk to others when you were feeling isolated	0.857	0.805	0.588
	2. You have used your mobile phone to talk to others when you were feeling lonely	0.863		
	3. You have used your mobile phone to make yourself feel better when you were feeling down	0.535		
Productivity loss	1. You find yourself occupied on your mobile phone when you should be doing other things, and it causes problems	0.783	0.738	0.585
	2. Your productivity has decreased as a direct result of the time you spend on the mobile phone	0.746		

FL = standardized factor loading; CR = composite reliability; AVE = average variance extracted.

**Table 3 ijerph-19-07578-t003:** Confirmatory factor analysis of the future anxiety scale.

	Item	FL	CR	AVE
Future anxiety scale	I am afraid that the problems which trouble me now will continue for a long time	0.795	0.897	0.638
	I am terrified by the thought that I might sometimes face life’s crises or difficulties	0.868		
	I am afraid that in the future my life will change for the worse	0.874		
	I am afraid that changes in the economic and political situation will threaten my future	0.692		
	I am disturbed by the thought that in the future I will not be able to realize my goals	0.749		

FL = standardized factor loading; CR = composite reliability; AVE = average variance extracted.

**Table 4 ijerph-19-07578-t004:** Confirmatory factor analysis of the child and adolescent mindfulness measure.

Dimension	Item	FL	CR	AVE
Awareness and non-judgment	1. I get upset with myself for having feelings that do not make sense	0.734	0.819	0.436
	2. At school, I walk from class to class without noticing what I am doing	0.524		
	3. I keep myself busy so I do not notice my thoughts or feelings	0.576		
	4. It is hard for me to pay attention to only one thing at a time	0.593		
	5. I think about things that happened in the past instead of thinking about things that are happening right now	0.742		
	6. I get upset with myself for having certain thoughts	0.751		
Acceptance	1. I tell myself that I should not feel the way I am feeling	0.723	0.798	0.502
	2. I push away thoughts that I do not like	0.552		
	3. I think that some of my feelings are bad and that I should not have them	0.820		
	4. I stop myself from having feelings that I do not like	0.711		

FL = standardized factor loading; CR = composite reliability; AVE = average variance extracted.

**Table 5 ijerph-19-07578-t005:** Discriminant validity.

DIMENSION	NCOV-CC	NCOV-TS	MPA-ICC	MPA-FAL	MPA-ES	MPA-PL	FA	CAM-ANJ	CAM-AC
NCOV-CC	** *0.759* **								
NCOV-TS	0.668 ***	** *0.739* **							
MPA-ICC	0.207 ***	0.230 ***	** *0.667* **						
MPA-FAL	0.132 ***	0.203 ***	0.590 ***	** *0.723* **					
MPA-ES	0.126 ***	0.124 ***	0.426 ***	0.491 ***	** *0.767* **				
MPA-PL	0.162 ***	0.158 ***	0.602 ***	0.472 ***	0.348 ***	** *0.765* **			
FA	0.510 ***	0.548 ***	0.354 ***	0.284 ***	0.256 ***	0.317 ***	** *0.799* **		
CAM-ANJ	−0.252 ***	−0.296 ***	−0.563 ***	−0.534 ***	−0.368 ***	−0.552 ***	−0.412 ***	** *0.660* **	
CAM-AC	−0.059	−0.060	−0.290 ***	−0.293 ***	−0.262 ***	−0.262 ***	−0.201 ***	0.349 ***	** *0.709* **
*M*	2.557	2.801	2.688	2.712	3.096	2.702	2.970	2.480	1.911
*SD*	0.912	0.884	0.778	0.912	0.923	0.858	1.319	0.706	0.757

N = 840. The bold and italic numbers in the diagonal are the square root of AVE. Numbers in the lower diagonal denote the correlation coefficients of two dimensions. *** *p* < 0.001; M = mean; SD = standard deviation; AVE = average variance extracted. NCOV, COVID-19 victimization experience; NCOV-CC, catastrophic cognition; NCOV-TS, trauma symptoms. MPA, mobile phone addiction; MPA-ICC, inability to control craving; MPA-FAL, feeling anxious and lost; MPA-ES, withdrawal/escape; MPA-PL, productivity loss. FA, future anxiety; CAM, child and adolescent mindfulness; CAM-ANJ, awareness and non-judgment; CAM-AC, acceptance.

**Table 6 ijerph-19-07578-t006:** Descriptive statistics and correlation analysis.

Variable	*M*	*SD*	COVID-19 Victimization Experience	Mobile Phone Addiction	Future Anxiety	Mindfulness
COVID-19 victimization experience	2.679	0.820	1			
Mobile phone addiction	2.769	0.690	0.240 ***	1		
Future anxiety	2.970	1.319	0.579 ***	0.382 ***	1	
Mindfulness	2.252	0.601	−0.244 ***	−0.625 ***	−0.392 ***	1

N = 840. *** *p* < 0.001. M = mean; SD = standard deviation.

**Table 7 ijerph-19-07578-t007:** Testing the mediation model of future anxiety.

Variable	Model 1 Mobile Phone Addiction			Model 2 Future Anxiety			Model 3 Mobile Phone Addiction		
	B	SE	95% CI	B	SE	95% CI	B	SE	95% CI
COVID-19 victimization experience	0.202 ***	0.028	(0.136, 0.268)	0.931 ***	0.045	(0.838, 1.023)	0.02	0.033	(−0.050, 0.095)
Future anxiety							0.191 ***	0.020	(0.149, 0.233)
*R*²	0.058			0.335			0.146		
*F*	51.228 ***			421.813 ***			71.684 ***		

B are unstandardized coefficients; SE, standard error; CI, confidence interval. *** *p* < 0.001.

**Table 8 ijerph-19-07578-t008:** Testing the moderated mediation model.

Variable	Model 1 Future Anxiety			Model 2 Mobile Phone Addiction		
	B	SE	95% CI	B	SE	95% CI
COVID-19 victimization experience	0.827 ***	0.044	(0.728, 0.920)	0.024	0.033	(−0.047, 0.097)
Mindfulness	−0.599 ***	0.061	(−0.722, −0.478)			
COVID-19 victimization experience × mindfulness	0.159 **	0.059	(0.045, 0.275)			
Future anxiety				0.191 *** 0.020 (0.149, 0.234)		
*R*²	0.407			0.146		
*F*	191.064 ***			71.684 ***		

B are unstandardized coefficients; SE, standard error; CI, confidence interval. ** *p* < 0.01, *** *p* < 0.001.

## Data Availability

The data presented in this study are available on request from the corresponding author.

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
