# Peer review of "COVID-19 Victimization Experience and College Students’ Mobile Phone Addiction: A Moderated Mediation Effect of Future Anxiety and Mindfulness"

_ijerph, 2022, doi:10.3390/ijerph19137578_

Round 1

Reviewer 1 Report

1.     The authors may rewrite the introduction part. For example, in the first paragraph, the authors only argue that factors leading to mobile phone addiction among college students must be explored. So where is the COVID-19 pandemic? Furthermore, why do we should study the mobile phone addiction of college students during the pandemic?

2.     It’s well done that the authors review relevant literature in the paragraph two. However, they did not point out the research gaps. What’s the contribution of the study?

3.     For part 2.2, the authors should focus on the relationship between COVID-19 victimization experience and mobile Addiction. The current version still includes some irrelevant content.

4.     In Table 2, the AVE of Inability to control craving is lower than 0.5. So why the authors suggest this is acceptable?

5.     The authors did not report the discriminant validity.

Author Response

Dear reviewer:

We really appreciate your time and effort in reviewing the manuscript and your comments are big helps in further improving the quality of the manuscript.

We have studied your comments carefully and revised the manuscript accordingly. Please find our answers point-by-point in blue font.

Reviewer 2 Report

My impression is that it has a lot of room for improvement. It is necessary to give it a scientific style and follow the rules of the magazine. It has to make many changes in the content of the article. Currently, as it is written, I do not suggest its publication. Many changes need to be made. I am enclosing some of my comments for its improvement.

Author Response

(The authors gave the same response as above.)

Reviewer 3 Report

The manuscript, entitled "COVID-19 Victimization Experience and College Students’ Mobile Phone Addiction: Mediating Role of Future Anxiety and Moderating Role of Mindfulness" is an interesting and timely work about possible predictors of COVID-19 pandemic consequences on mobile phone addiction among college students in China. Although the paper has a merit, many parts of the manuscript should be improved before reconsidering of it for publication in the journal.

1. Introduction should finish the expected mediation model for verification, after presentation of hypotheses. Please move the whole section 3.1. Research Framework from the Methods section into Introduction.

2. Methos section is chaotic. Procedure of research should be described in a separate section, including all relevant information about the data collection, sampling method, and ethical questions related to the study. How were online questionnaires distributed among college students? Via social media or university mailing list, or another way? Was anonymous and voluntary the study participation? Was informed consent presented before participation? How many students refused participation in the study? What about ethic IRB approval? Unfortunately, replication of this study os impossible without all these missing information.

3. Statistical analysis section should include all statistical methods used in the study, information, what specific hypothesis is verified using each method, what dependent and independent variables are examined in each model, etc. What type of bootstrap technique is used, and how many sampling was implemented?

4. Please use standard statistical symbols, like beta greak symbol instead of SFL.

5. Convergent and discriminant analysis (including Fornell and Larcker criterion and HTMT) should be performed in the results section (instead of the Methods), as a preliminary analysis to check parametric properties of the data. In addition, one complex analysis should be presented instead of several tests for particular measures. 

6. Interpretation of the convergent analysis is inappropriate. Authors refer to Fornell and Larcker [78] that they suggest AVE > 0.36 as acceptable. Unfortunately, I have not found in the referred publication [78] any information supported this assumption. Instead, please see the current publication that suggest AVE > 0.5, beta > 0.5, CR > 0.7, and correlation r < 0.7, as acceptable levels:

Cheung, G. W., & Wang, C. (2017). Current Approaches for Assessing Convergent and Discriminant Validity with SEM: Issues and Solutions. In G. Atinc (Ed.), Academy of Management Proceedings, Vol. 2017 (No. 1), https://doi.org/10.5465/ambpp.2017.12706abstract

7. Similarly, model fit indices are wrongly interpreted, inconsistently with the referred literature (McDonald & Ho, 2002) and current standards. One of the most important statistics is Chi-square test, but this is missing in the study. The other statistics, like CFI, TLI, etc., are wrongly interpreted. Please see the current standards for cut-off criteria of goodness of fit. See for example the webpage: https://easystats.github.io/effectsize/reference/interpret_gfi.html

8. The tables 6 and 7 (with mediation and moderated mediation models) are unclear, and do not include important statistics. Please see the APA style guideline to learn what statistics should be reported and how the data should be arranged in a table. 

9. The limitations section should be better described, including more problems, like sampling method, web-based questionnaire as a tool, low AVE scores in several measures, etc.

Author Response

(The authors gave the same response as above.)

Round 2

Reviewer 1 Report

It's well enough to be published.

Reviewer 2 Report

Congratulations. The paper has improved significantly.

Reviewer 3 Report

The Authors improved the manuscript substantially. I recommend the paper for publication in the IJERPH.